# Effects of Age Stereotypes of Older Workers on Job Performance and Intergenerational Knowledge Transfer Intention and Mediating Mechanisms

**DOI:** 10.3390/bs14060503

**Published:** 2024-06-17

**Authors:** Ying Wang, Weiwei Shi

**Affiliations:** School of Government, Beijing Normal University, Beijing 100875, China; xjtuwy@bnu.edu.cn

**Keywords:** older workers, age stereotype, intergenerational knowledge transfer intention, successful aging at work, age discrimination

## Abstract

The workforce is aging with the population aging. How to effectively manage and motivate older workers is significant for elderly human resources development and the sustainable development of enterprises in organizations. Age stereotypes of older workers refer to people’s beliefs and expectations about a specific group of 45–65-year-olds in the workplace. This paper examines the effect of age stereotypes of older workers on job performance and intergenerational knowledge transfer intention. This study carried out two research designs, a questionnaire survey and an experimental study, to explore the effects of positive and negative age stereotypes of older workers on job performance and intergenerational knowledge transfer intention within an organizational context and underlying mediating mechanisms. The results showed that positive stereotypes of older workers significantly positively affected job performance and intergenerational knowledge transfer intention. In comparison, negative stereotypes of older workers significantly negatively affected job performance and intergenerational knowledge transfer intention, and self-perception of aging substantially mediates the effects. This study broadens the research field on the impact of positive and negative age stereotypes on older workers in organizational contexts. This study guides organizations in reducing age discrimination, creating an inclusive workplace environment, and achieving the successful aging of older workers.

## 1. Introduction

Unprecedented demographic changes as a result of increased longevity and extremely low birth and death rates have led to the ageing of the global population. As the most populous country in the world, the increasing elderly population in China is also a pressing global issue. According to the World Population Outlook 2022 report by the United Nations, it is projected that by 2050, over 500 million Chinese individuals will be 60 years old or older, accounting for 38.81% of the population. As China gradually raises the retirement age, the number of experienced and skilled personnel in the workforce is expected to increase. However, research has emphasized that it is becoming increasingly challenging for organizations to effectively engage senior staff and leverage the knowledge and expertise of older employees [1,2,3].

Age stereotypes refer to the beliefs and expectations of older individuals as a specific social group [4], which can be divided into positive age stereotypes and negative age stereotypes. On the one hand, older workers are believed to have accumulated more professional skills and practical knowledge and to have a higher ability to predict and solve work problems, thus not only maintaining but also promoting work performance [5]; on the other hand, older employees are considered to be resistant to change, unwilling to learn new knowledge, and less productive [6]. Stereotypes, prejudices, and discriminatory behaviors against older employees exist in various organizations [7].

Employers and employees generally hold negative age stereotypes about older workers in the workplace. In particular, managers often hold negative age stereotypes, namely, that older workers are less flexible, adaptable, or productive [5]. Many empirical studies have proved that negative age stereotypes of managers toward older workers may lead to discrimination, including fewer hiring chances and fewer learning, development, and promotion opportunities [5,8,9,10]. This discrimination by managers can affect older employees’ work performance and engagement [11]. However, older workers who are a stereotyped group tend to hold age stereotypes themselves [12]. Whether and how negative age stereotypes held by older workers affect individual outcomes, such as work performance and motivation, has been less frequently addressed. In addition, current research has found that for organizations to develop human resources for older workers and promote the successful aging of older workers, older workers must be motivated to actively maintain a high level of ability and motivation to continue working [2]. Therefore, managers’ stereotypes of older workers and related work-related outcomes are significant but not the focus of this study. From the older employees themselves, it is crucial to explore the influence of age stereotypes held by older employees on individual work motivation and work outcomes.

Current research on age stereotypes has mainly focused on the field of geriatric psychology, exploring the impact of age stereotypes on the physiology, cognition, and behavior of older people [4,13,14,15]. Some studies have introduced age stereotypes into organizational contexts to explore the impact of age stereotypes on older employees, focusing on the impact of negative age stereotypes on learning and development motivation [16], retirement intention [17], job satisfaction [18], and job engagement [19]. However, the impacts of positive age stereotypes on older workers have been ignored [20]. Since positive and negative age stereotypes may exist simultaneously, and their effects on the work motivation and attitude of older employees are different, it is essential to consider both positive and negative age stereotypes’ impact on work outcomes for older employees to reduce age discrimination, promote successful aging, and maximize the organization’s human resources.

Older employees’ work performance is the focus of concern for organizations, and age-induced discrimination largely stems from questioning the work performance of more senior employees. However, the impact of age stereotypes on employee performance is rarely debated [21,22]. Apart from job performance, there are a few studies on the quality of intergenerational contact. As the new generation of employees accelerates to enter the workplace and the number of older employees increases, the intergenerational differences within organizations widen [23]. Older employees have rich skills and experience and may be willing to invest resources in intergenerational knowledge transfer with emotional significance out of the pursuit of life meaning [24]. However, age-related discrimination and negative age stereotypes will further affect the awareness of age differentiation and age anxiety within organizations. Therefore, it is worth exploring whether different age stereotypes subtly affect older employees’ intergenerational knowledge transfer intention.

Based on this, the marginal contribution of this paper lies in the following aspects: First, this study broadens the research on the mechanism of the influence of age stereotypes on the intergenerational knowledge transfer intention of older employees’ work performance within organizational contexts and also extends empirical research into the consequences of age discrimination on older employees. Second, this study offers a new viewpoint by examining the impact of positive and negative age stereotypes in organizations, moving beyond the traditional emphasis on negative age stereotypes. This shift in emphasis improves understanding of the elements that support older workers’ successful aging in the workplace. Finally, this study enriches the antecedents of older employees’ job performance and intergenerational knowledge transfer intention.

To this end, this study uses both the questionnaire method and experimental research. It combines explicit and implicit priming to conduct research design to jointly verify the effects of positive and negative age stereotypes on work performance, the intergenerational knowledge transfer intention of older employees, and the mediating role of self-perceived aging. Individuals hold positive and negative age stereotypes [25]. Study 1 measures the positive and negative age stereotypes of elderly employees at two dimensions through the scale, and this external display activation mode can reflect the long-term age stereotypes held by elderly employees. By referring to the manipulation methods of previous experimental methods, study 2 carries out the activation manipulation of different titers on the variable of age stereotypes. It implicitly awakens positive or negative stereotypes of older employees in their minds, focusing on the immediate effects after activating the age stereotypes. Through the mutual verification of these two methods, this study further explores the impact of age stereotypes on older employees’ work in organizations (Figure 1).

## 2. Literature Review and Hypotheses

### 2.1. Definition of Older Employees

There is no single standard for defining the age of older workers. Based on career development theories and empirical studies, the age range of “older workers” is generally considered 45–65. First, according to career development stage theory, there are different career development processes and career needs and goals in different life stages. These stages include the growth, exploration, establishment, maintenance, and decline periods [26,27]. Among them, individuals aged 45–65 years are in the career maintenance period, during which the main development task is to maintain professional status and achievement, during which they may face occupational fatigue or need to re-evaluate career goals, seek new development opportunities, maintain family and work balance, and plan for retirement. Second, this paper references a few empirical studies that focus on older workers, such as Kulik et al.’s study published in the *Academy of Management Journal* in 2016 and the study on the impact of the threat of age stereotypes on the work engagement of older workers [28]. A study published in the *Journal of Applied Psychology* on the intergenerational knowledge transfer of older employees and the impact of personality traits and job reshaping on successful workplace aging [24] defined the age of older employees as 45 years old.

Setting the age of older workers at 65 is mainly related to China’s retirement policy and China’s labor force participation rate. While the retirement age is generally over 65 in the world’s major economies, China still uses a 1978 retirement policy. The legal retirement age is 50 years for female workers, 55 years for female cadres, and 60 years for male cadres and workers [29]. China is implementing the Gradual Delay Retirement Policy, which is expected to be delayed to 65 years old by 2045. In addition, according to the 2016 China Urban Labor Survey (CULS) data, the labor participation rate of the elderly (aged 65 and above) is only 1.8%, indicating that the number of over-65s in the workforce is particularly low [30]. Therefore, based on the context, the upper age limit for older workers is 65.

### 2.2. Age Stereotypes

Stereotypes refer to the stereotyping of group impressions, and age stereotypes refer to people’s beliefs and expectations about the elderly as a specific social group [4]. According to the evaluation object, age stereotypes can be divided into group age stereotypes and self-stereotypes. The group age stereotypes refer to the stereotype of the entire elderly group, and self-stereotyping refers to stereotyping oneself as an elderly person [14,31]. According to the content of age stereotypes, there are two types of age stereotypes: positive and negative. Positive stereotypes regard elderly people as healthy, wise, and energetic, while negative stereotypes consider them as old, lonely, and helpless [25]. In addition, age stereotypes can be divided into implicit and explicit age stereotypes according to whether they can be realized [15,32]. Implicit age stereotypes are those that are activated subliminally through subliminal priming, while explicit age stereotypes are activated through suprathreshold priming. Research has shown that the implicit age stereotypes that people hold can be more harmful than explicit age stereotypes and can significantly impact individuals [14].

Previous research has also found that employers’ stereotypes about older workers’ productivity reflect hard and soft skills. Soft skills are “organizational citizenship behavior,” including commitment to the organization, reliability, and social skills. In contrast, hard skills include flexibility, physical and mental capacity, and the willingness to learn new technology skills [33,34,35]. The comparative advantage of older workers primarily lies in their soft qualities, but in evaluating the productivity of older and younger employees, hard skills are considered much more important than soft qualities [34,35]. This means that attitudes and stereotypes toward older employees are intricate, as older individuals are perceived to have both positive and negative attributes. Positive characteristics mainly focus on evaluations of hard skills, while negative attributes primarily center on assessments of hard skills [34,35]. Although this categorization method differs slightly from positive and negative age stereotypes, as positive and negative stereotypes emphasize overall perceptions and expectations of older employees, not just work skills, both views fundamentally suggest that evaluations of older employees should encompass both positive and negative aspects. Therefore, even though there may be differences in the criteria used to assess soft and hard skills, these differences are unlikely to impact the primary findings of our study significantly.

### 2.3. Age Stereotypes and Job Performance

Based on the resource conservation theory, resources are defined as anything that an individual perceives can help them achieve their goals, including material resources (such as plants), condition resources (such as seniority and position), individual characteristic resources (such as self-efficacy, IQ), and ability resources (such as time and knowledge). Individuals strive to acquire, maintain, cultivate, and protect their cherished resources. When individuals face the possible loss or actual loss of resources, self-pressure will be generated. At this time, individuals will take action to reduce resource loss to maintain existing resources, and the impact of individual resource loss is far more significant and longer than that of resource acquisition [36,37].

Based on the resource acquisition process, older employees with positive age stereotypes will think that older employees have rich work experience and skills, are loyal, and are trustworthy [5]. This positive self-image affirms their workability and self-worth and helps them gain the support and trust of their colleagues, leaders, and the organization [38]. In addition, older employees who hold positive age stereotypes will attach importance to the training and development of their professional knowledge and skills and acquire additional resources through continuous learning, thus continuously experiencing the “value-added resources spiral”.

Regarding the resource loss process, older employees who hold negative age stereotypes will believe that older employees have no value, lack ability, and cannot learn and master new skills [33]. On the one hand, elderly employees are in a disadvantageous position in the organization and feel the lack of organizational support, resulting in high anxiety and high pressure perception, which requires the elderly employees to continuously consume time and attention to cope with negative emotions and suffer continuous consumption of individual resources [36], so they are reluctant to devote too much energy to improve work performance. On the other hand, the negative age stereotypes held by older employees may mean that in the management practice of the organization, more senior employees cannot acquire the training and development resources needed to improve work skills and solve work problems [33]. The organization cannot improve its work skills when the resources are continuously consumed and cannot be replenished in time. Older workers take a defensive stance, reducing their commitment to work to avoid further losses of their resources.

The previous empirical study found the effects of age stereotypes on the work outcomes of older workers in the workplace [16,17,19,33,39,40]. In the workplace, older employees are often given negative stereotypes, i.e., they are considered resistant to change, unwilling to participate in training and development, and not easily motivated [33]. In 2008, Maurer et al. found that negative age stereotypes of older employees negatively affect their self-efficacy and career development motivation [16]. Positive age stereotypes reduce the intention of older workers to retire early [17,39]; age stereotype threats caused by negative age stereotypes will reduce the work engagement and performance of older employees [19,40].

Based on the above analysis, the following hypothesis is proposed:

**Hypothesis** **1a:**
*The positive age stereotypes of elderly employees have a significant positive impact on job performance.*


**Hypothesis** **1b:**
*The negative age stereotypes of elderly employees have a significant negative impact on job performance.*


### 2.4. The Age Stereotypes and Intergenerational Knowledge Transfer Intention

Intergenerational knowledge transfer refers to transferring the experience and knowledge of older employees to younger employees, which prevents organizational forgetfulness, fosters the creation of new knowledge, and ensures the continuity of organizational knowledge [41]. Intergenerational knowledge transfer intention refers to the intention of different intergenerational knowledge transfer behaviors, which is a prerequisite for initiating intergenerational knowledge transfer activities [23]. For elderly employees, achieving intergenerational knowledge transfer is conducive to regaining pride and confidence in their work [42], satisfying emotional needs and nurturing values, promoting a sense of age identity, and feeling valued and respected [23]. However, knowledge is a scarce resource within an organization. To maintain their core competence in the organization, older employees may also protect tacit knowledge, such as solving complex work problems and interpersonal skills, which may diminish their intention to share knowledge between generations [23].

The resource conservation theory suggests that knowledge is a valuable resource within an organization that can be transferred, imitated, and irreplaceable [43]. On the one hand, older employees hold a positive stereotype that older employees are knowledgeable, experienced, willing to share, and valuable to the organization. Older employees are often viewed positively as knowledgeable, experienced, and willing to share their expertise. While sharing knowledge with younger employees may require older employees to invest their time, energy, and other resources, the recognition and affirmation of the value of their work by the organization and their colleagues can activate internal motivation among elderly employees [44]. This positive information will help individuals replenish lost resources promptly. Thus, older employees will have more intergenerational knowledge transfer intention to maintain their positive image in the organization. On the other hand, regarding resource depletion, older employees tend to hold negative age stereotypes towards older groups in the organization, assuming that they lack flexibility, adaptability, and productivity. Transferring knowledge from older employees to younger ones can be challenging. It demands their time and energy and puts their workplace status at risk. Additionally, they may have to deal with negative evaluations and put in extra effort to restore their self-esteem to avoid losing more resources [36]. As a result, these factors limit the resources older workers are willing to invest in intergenerational knowledge transfer.

Previous research indicates that older employees who find and pursue meaning in life are more likely to have an intention to transfer knowledge to younger generations [44]. Trust and respect among older workers foster a perception that they can share knowledge with others [23]. However, older workers often hold negative stereotypes about aging, believing that the organization does not support them and that they are mistreated. An appropriate organizational environment is necessary to influence older employees’ intention to transfer knowledge to younger generations [44]. The perceived age discrimination in the organization significantly reduces older employees’ intention to contribute their knowledge to younger generations [45].

Based on the above analysis, the following hypothesis is proposed:

**Hypothesis** **2a:**
*The positive age stereotypes of elderly employees significantly positively influence the intergenerational knowledge transfer intention.*


**Hypothesis** **2b:**
*The negative age stereotypes of elderly employees significantly negatively affect the intergenerational knowledge transfer intention.*


### 2.5. The Mediating Effect of Self-Perception of Aging

Self-perception of aging (SPA) refers to the subjective perception and emotional response of elderly individuals when they confront the aging process in their physiology, psychology, and society [46]. It is also an individual’s perception or view of their mental state formed through observing their aging [47].

Age stereotypes and self-perception of aging are two different types of aging views that interact with each other and influence the outcome of aging [48]. Age stereotypes are defined as socially shared beliefs about aging and older people as a group [49], representing attitudes about others’ aging and what it means to be an older person [50]. In contrast to general age stereotypes, self-perceptions of aging refer to individuals’ experiences of growing older [51]. Thus, whereas age stereotypes target older adults as a group strongly influenced by societal views on aging, self-perception of aging reflects a person’s own behavioral experiences of growing older and targets the aging self.

How do two different views on aging, age stereotypes, and self-perception of aging interact and influence aging outcomes? Some studies have theorized that age stereotypes precede self-perceived aging [52]. According to stereotype embedding theory, age stereotypes held throughout a lifetime become internalized, thereby influencing the development of personal views on aging [52]. Age stereotypes are internalized by individuals at a young age and play a significant role in the cognitive representation of the self in old age. This leads to the assimilation effect of an individual’s self-view and behavior with their internalized age stereotype [53,54], which can result in positive or negative self-perceptions of aging.

Self-categorization theory suggests that self-perception is influenced by how people categorize themselves, attribute in-group characteristics to themselves, and self-stereotype. The theory proposes that individuals integrate external social categories into their inner selves through group identification, leading to self-perception and behavior more consistent with in-group norms and stereotypes [55]. Research suggests that older workers who maintain a positive outlook toward their peers tend to exhibit a sense of responsibility, a can-do attitude, and a wealth of valuable knowledge and experience. Such positive self-perceptions contribute to higher levels of career self-efficacy and self-esteem, leading to a more optimistic outlook toward their aging process [38]. In contrast, older workers who harbor negative stereotypes about aging tend to regard it as a time of low efficiency and a lack of innovation. This self-perception can seriously impact their psychological well-being, leading to negative emotions such as anxiety and insecurity. As a result, they may develop a negative attitude toward the aging process, which can negatively affect their interactions with other organization members [56].

Previous studies have also shown a correlation between age stereotypes and self-perceived aging [57]. Age stereotypes are a crucial factor affecting self-perceived aging, and individuals with positive perceptions of the elderly will also internalize positive self-perception of aging [52]. Self-perceived aging further influences successful age-related outcomes such as health, cognitive ability, personal expectations, and subjective well-being in old age [31,54]. The negative identity of older employees induced by the age stereotypes threat caused by negative stereotypes reduces their work involvement [28] and their self-efficacy and thus their work performance [19,40].

Based on the above analysis, the following hypothesis is proposed:

**Hypothesis** **3a:**
*Self-perceived aging mediates the relationship between positive age stereotypes influence on job performance and intergenerational knowledge transfer intention.*


**Hypothesis** **3b:**
*Self-perceived aging mediates the relationship between negative age stereotypes’ influence on job performance and intergenerational knowledge transfer intention.*


This study employed two research designs to test the hypotheses proposed: questionnaire and experimental research. Study 1 primarily relied on the self-assessment of older employees to measure positive and negative age stereotypes, self-perceived aging, job performance, and intergenerational knowledge transfer intention via scales for initial validation of the model. Study 2 used the experimental research method to further validate the first study by exploring the effects on the innovation dimension of job performance and intention to transfer knowledge between generations through the implicit manipulation of activating positive or negative age stereotypes in older employees.

## 3. Study 1: Survey

The first study aims to test the whole research hypothesis and verify the mediating relationship between positive and negative age stereotypes and the effect of self-perceived aging on job performance and intergenerational knowledge transfer intention. The samples were collected through self-evaluation of the questionnaire, and the data were analyzed by SPSS and AMOS software.

### 3.1. Sample and Procedure

The data used in this study were collected from questionnaires issued and collected between August 2022 and March 2023. The sample sources were public sector and enterprise employees. The samples were obtained from government departments and public institutions of Yuxian County, Yangquan City, Shanxi Province, for public sector employees. The research team established contact with the personnel department of each unit and obtained the list of voluntary participants in the questionnaire survey with the consent of the unit’s person in charge. The research team distributed offline questionnaires to 230 working elderly employees over 45 years old and recovered 201 questionnaires, yielding a recovery rate of 87.39%. For enterprise employees, the samples were randomly selected from private and state-owned enterprises across the country through the online research platform Credamo. Two hundred twenty employees over 45 years old were distributed online questionnaires, and 200 questionnaires were collected, resulting in a recovery rate of 90.91%. After two rounds of questionnaire recovery, the research team finally obtained 401 valid questionnaires, with an effective recovery rate of 89.11%. All senior employees who participated in the survey read the informed consent form before responding and voluntarily decided to participate. After completing the questionnaire, participants received a reward of 20 RMB.

The sample age ranged from 45 to 65 years old, with a mean age of 50.8 years (SD = 4.75). There are 192 men and 209 women, 50% from the non-public sector (government departments and public institutions), 50% from enterprises (state-owned enterprises and private enterprises), 64.84% from management, and 35.16% from non-management. The average working life is 25.5 years, the longest is 43 years, and the shortest is 10 years. From the perspective of education level, 51.12% are below the undergraduate level, and 48.88% are above the undergraduate level. In terms of health, 14.46% of workers think that their health is poor, and 85.54% think that their health is good.

### 3.2. Measures

#### 3.2.1. Age Stereotypes

Positive age stereotypes and negative age stereotypes were measured by Levy’s (2004) scale, with seven items each [25]. The scale was modified to assess an individual’s attitudes and expectations towards older workers and consisted of 14 items: seven positives (active, wise, patient, capable, healthy, creative, independent) and seven negatives (inflexible, senile, grumpy, incompetent, sick, rigid, dependent). Positive age stereotypes include “Do you think older workers are patient?” and “Do you think older workers are creative?” Items of negative age stereotypes include “Do you think older workers are confused?” and “Do you think older workers are old?” The scale uses Likert’s 4-point scale (1 = strongly disagree; 4 = strongly agree). The higher Cronbach’s was 0.87. The higher the negative age stereotypes score, the more negative the age stereotypes of older employees. In this study, Cronbach’s alpha was 0.93.

#### 3.2.2. Self-Perceived Aging

Using a validated China version, self-perceived aging was measured via the Hu and Liu (2018) scale [58]. The scale’s validity has been widely verified in factor structure and convergence validity. The scale consists of 17 items and 5 dimensions: negative outcome and control, positive outcome, chronic time, positive control, and affective representation. Example items were “I always put myself in the older category” and “Getting older makes me more dependent”. The scale uses a 5-point Likert scale (1 = strongly disagree, 5 = strongly agree), and questions 4, 5, 6, 8, 9, and 10 are scored in reverse. Cronbach’s alpha was 0.94.

#### 3.2.3. Job Performance

Using a validated China version, job performance was measured with 12 items adapted from the Han et al. (2007) scale [59]. The scale’s validity has been widely verified in factor structure and convergence validity. The four dimensions are task performance, relationship performance, learning performance, and innovation performance. Example items were “I can better perform my current job responsibilities through learning” and “I can turn innovative ideas into practical applications”. The scale uses a 5-point Likert scale. The higher the total score, the higher the older worker’s performance. Cronbach’s alpha was 0.91.

#### 3.2.4. Intergenerational Knowledge Transfer Intention

Using a validated China version, the intergenerational knowledge Transfer Intention Scale was measured with five items adapted from Wang and Zuo [60]. The sample items are “When young colleagues ask me for advice, I will be happy to tell them” and “I will often provide my work manual, methods, and models to young colleagues”. The scale uses a 5-point Likert scale. The higher the total score, the more inclined to intergenerational knowledge transfer intention. Cronbach’s alpha was 0.83.

#### 3.2.5. Control Variables

Gender (1 = male; 2 = female), age, organization (1 = public sector; 2 = private sector), occupation (1 = manager; 2 = non-manager), tenure, education (1 = vocational/technical school; 2 = high school; 3 = junior college; 4 = bachelor’s degree; 5 = master’s degree), health (1 = very good; 2 = good; 3 = general; 4 = bad; 5 = very bad) were applied as control variables in the analysis, as previous studies have demonstrated that these variables may correlate with the older worker’s outcomes.

### 3.3. Analytical Strategy

SPSS 21.0 and Amos 23.0 were used for data analysis to test the theoretical model and research hypotheses. In this study, AMOS 23.0 was used for confirmatory factor analysis to demonstrate construct validity. SPSS 21.0 was used for scale reliability analysis and common method bias analysis, correlation analysis, regression analysis, and mediation testing were used to conduct statistical analysis of the data, aiming to explore the effects of age stereotypes and self-perceived aging on job performance and intergenerational knowledge sharing willingness of older employees in organizational contexts and their internal mechanisms.

### 3.4. Research Results

#### 3.4.1. Confirmatory Factor Analysis

In this study, AMOS software was used to conduct confirmatory factor analysis, and the test results are shown in Table 1. Compared with other competitive models, the five-factor model was assumed to have the most ideal fitting effect (χ^2^ = 912.925, df = 339, χ^2^/df = 2.693, CFI = 0916, IFI = 0.917, TLI = 0.908, GFI = 0.842, RMSEA = 0.065), indicating that these six variables represent different constructs.

A series of confirmatory factor analyses (CFA) was conducted to assess the discriminability of the study variables. As shown in Table 1, compared with other competitive models, the five-factor model was assumed to have the most ideal fitting effect (χ^2^ = 912.925; df = 339; χ^2^/df = 2.693; CFI = 0916; IFI = 0.917; TLI = 0.908; GFI = 0.842; RMSEA = 0.065), which shows that the five study variables (i.e., positive and negative age stereotypes, self-perception of aging, job performance, and intergenerational knowledge transfer intention) in this study have good discriminant validity.

#### 3.4.2. Testing Common Method Bias

To control the common method bias problem, the Harman single-factor test was used to perform the standard method bias test [61], which included the five core study variables and all control variables. The results showed that the unrotated first factor explained 29.240% of the total variation and did not account for 40%.

#### 3.4.3. Descriptive Statistics and Correlations

Table 2 shows the descriptive statistics of the study variables and the correlation coefficient matrix. Positive age stereotypes were significantly negatively correlated with self-perceived aging (r = −0.604; *p* < 0.01), positively correlated with job performance satisfaction (r = 0.595; *p* < 0.01), and positively correlated with intergenerational knowledge transfer intention satisfaction (r = 0.510; *p* < 0.01). Negative age stereotypes were positively correlated with self-perceived aging (r = 0.603; *p* < 0.01), negatively correlated with job performance (r = −0.383; *p* < 0.01), and negatively correlated with intergenerational knowledge-sharing intention (r = −0.289; *p* < 0.01). Self-perceived aging was negatively correlated with job performance (r = −0.542; *p* < 0.01) and with intergenerational knowledge transfer intention (r = −0.511; *p* < 0.01). The above correlation statistics provide a preliminary basis for the subsequent hypothesis testing.

### 3.5. Hypothesis Testing

#### 3.5.1. Main Effect Test

To verify hypotheses 1a, 1b, 2a, and 2b, this study uses SPSS 21.0 as a data processing tool and adopts hierarchical regression analysis to test the hypothesis. First, job performance and standby knowledge transfer were set as dependent variables, and seven control variables, such as gender, age, working years, unit type, job type, education level, and health status, were regression performed to obtain model 3 and model 6, respectively. Then, positive and negative age stereotypes are added to obtain model 4 and model 7, respectively, and the specific results are shown in Table 3. According to the results of model 4, the positive age stereotypes has a significant positive impact on job performance (β = 0.415; *p* < 0.001), and the negative age stereotypes has a significant negative impact on job performance (β = −0.073; *p* < 0.05). Hypotheses 1a and 1b are verified. According to the results of model 7, the positive age stereotypes significantly positively affected intergenerational knowledge transfer intention (β = 0.7; *p* < 0.001), and the negative age stereotypes significantly negatively affected intergenerational knowledge transfer intention (β = −0.115; *p* < 0.05). Let us say that 2a and 2b are verified.

#### 3.5.2. Mediating Effect Test

This study used stepwise and bootstrapping methods to verify the mediating effect of self-perceived aging. The results of regression analysis are shown in Table 3. Based on introducing seven control variables, self-perceived aging was regressed, and Model 2 was obtained. Positive age stereotypes had a significant negative effect on self-perceived aging (β = −0.456; *p* < 0.001), and negative age stereotypes had a significant positive effect on self-perceived aging (β = 0.391; *p* < 0.001). Hypotheses 3a and 3b were partially verified.

Based on Model 4, taking job performance as the dependent variable and adding self-perceived aging, Model 5 is obtained. The results showed that when self-perceived aging was added, the positive age stereotypes significantly affected job performance (β = 0.333, *p* < 0.001) but the effect became smaller. Negative age stereotypes have no significant effect on job performance. Based on model 7, taking intergenerational knowledge transfer intention as the dependent variable and adding self-perceived aging, Model 8 is obtained. The results showed that when self-perceived aging was added, the positive age stereotypes had a significant positive effect on job performance (β = 0.521; *p* < 0.001), and the negative age stereotypes had no significant effect on intergenerational knowledge transfer intention. Combined with the above Hypotheses 1a, 1b, 2a, and 2b, it can be preliminarily determined that self-perceived aging mediates job performance and intergenerational knowledge transfer intention in both positive age stereotypes and negative stereotypes. Let us say that 3a and 3b are verified.

Then, bootstrapping was used to sample 5000 times for analysis repeatedly, and the results showed that the intermediary effect of self-perceived aging on job performance through positive stereotypes had a 95% confidence interval of [0.0485, 0.1207]. The mediating effect of self-perceived aging on intergenerational knowledge transfer intention through positive stereotypes amounted to a 95% confidence interval of [0.1133, 0.2571]. The mediating effect of self-perceived aging on job performance through negative stereotypes has a 95% confidence interval of [−0.1017, −0.0425]. The 95% confidence interval for the mediating effect of self-perceived aging on intergenerational knowledge transfer intention through negative stereotypes is [−0.2151, −0.099], and 0 is excluded in all four intervals. Therefore, self-perceived aging has a significant mediating effect on both positive and negative stereotypes of job performance and intergenerational knowledge transfer intention. So, 3a and 3b are verified.

## 4. Study 2: Experimental Study

First, by implicitly manipulating the experimental conditions of positive or negative age stereotypes, we explored the causal relationship between age stereotypes influence on job performance and intergenerational knowledge transfer intention. Second, study 2 focuses on the research of innovation performance in work performance. Innovation performance is significant to the future era of scientific and technological change, and population aging is an inevitable trend in China. Whether aging will affect people’s innovative ability and spirit and then affect scientific and technological innovation is an important issue concerned by the whole society. Therefore, in study 2, innovation performance was mainly studied through experimental tasks to explore the relationship between different age stereotypes and innovation performance.

### 4.1. Research Samples

In this experiment, a total of 63 working elderly employees aged 45–65 years old were recruited by local volunteer groups in Yangquan City, Shanxi Province, in July 2022. Because three subjects left the experiment midway for personal reasons, 60 effective participants were finally obtained. The average age of the participants was 51.13 years old (SD = 4.204), and the average working life was 30.63 years (SD = 4.801). Regarding organizations, 51.7% were from the public sector, and 48.3% were from the private sector. The distribution of educational background was 56.7% for less than a bachelor’s degree and 43.3% for a bachelor’s degree or higher. In terms of health, the overall health status is good; as many as 78.3% of the workers think their health is good, and 21.7% think their health is poor.

### 4.2. Experimental Material

In study 2, an aging stereotype lexeme applicable to the Chinese context was generated by issuing questionnaires. The aging stereotype thesaurus used in empirical research is from the English scale (from a non-Chinese context). To make the research results more convincing, the vocabulary used in this study’s “implicit manipulation activation” stage is from the “Chinese context aging stereotype thesaurus”. The sample of thesaurus comes from elderly working employees in China. A total of 400 questionnaires were sent out through research, and 369 were effectively recovered. The questionnaire consisted of two questions: “First, choose 12 words from the following 26 words that better fit your impression of the positive image of older workers (over 45 years old) in the workplace (e.g., sociable, leadership, insightful, intelligent, experienced, etc.)”; the second is “Choose 12 words from the following 26 words that better fit your negative image of older workers (over 45 years old) in the workplace (stubborn, dependent, inefficient, conservative, slow, etc.)”. According to the results of the questionnaire, the localized stereotype lexicon was obtained, including 12 positive and 12 negative words. The lexicon would be used as activation words for the subsequent implicit manipulation of the experiment.

### 4.3. Experimental Procedures

Study 2 conducted the experiment in person in a controlled laboratory environment. Prior to the experiment, participants selected convenient time slots for their participation. Based on these selections, the researcher divided the 60 participants into five groups of 12 individuals each. All experimental sessions were held in a designated meeting room provided by the volunteers. The researcher was present throughout the sessions, responded to participants, and provided necessary assistance to ensure that participants understood and completed the tasks accurately. Before the start of the experiment, the researcher explained the experiment’s purpose, procedure, and precautions to the participants and ensured that all participants signed informed consent forms.

All participants were randomly assigned to a positive older worker stereotype experimental group and a negative older worker stereotype control group, with 30 participants in each group. First, the subjects were subjected to 30 s positive aging stereotype or negative aging stereotype pre-activation manipulation and then to formal manipulation. Second, all subjects completed the age stereotype and self-perceived aging questionnaire. Then, a second stereotype activation manipulation was performed after completing the questionnaire. Then, after the experimental manipulation, the subjects completed the experimental situational tasks to test job performance and intergenerational knowledge transfer intention and filled in the questionnaire related to control variables. Finally, the researcher thanked the participants for their participation, explained the purpose of the experiment, and rewarded them with a corresponding voucher.

### 4.4. Experimental Manipulation and Measures

The first step is an implicit activation manipulation and manipulation activation test. Implicit activation manipulation effectively distinguishes between positive and negative groups by activating different titers of different subjects. Older workers stereotype positive and negative activation manipulation using up and down arrow word flash judgment task [4]. First, a detection stimulus “+” (1000 ms) is presented in the center of the computer screen; then, the target word is randomly presented above or below the “+”. The presentation time is between 17 ms and 50 ms. The subjects must judge whether the target word is above or below the “+” by pressing the up and down arrow. The experimental materials included positive and negative stereotype vocabulary. A localized stereotype lexicon was adopted. All words in the lexicon needed to be randomly repeated twice. The positive group included 2 category words (older worker, older comrade), 12 positive adjectives (experienced, wise, kind, erudite, respected, leading, insightful, warm-hearted, patient, dignified, tolerant), 4 neutral words (rushed, simple, high, every day), and 2 random repetitions. The negative group also included 2 category words (older worker, older comrade), 12 negative adjectives (old, lazy, complaining, rigid, passive, old-fashioned, inefficient, stubborn, outdated, ill-tempered, easily tired, careless), and 4 neutral words (rushed, simple, high, daily).

After the experimental manipulation, the effectiveness of the aging stereotype activation manipulation was tested by filling in the aging stereotype questionnaire. The positive stereotype groups scored significantly higher than the negative ones, indicating successful experiment manipulation. In the same study, Levy’s (2004) scale was used in the aging stereotype questionnaire [25], which showed good consistency (Cronbach’s α = 0.88).

#### 4.4.1. Self-Perceived Aging

The second step was to complete the self-perceived aging questionnaire after the experimental manipulation. Self-perceived aging questionnaires were completed immediately after the participants were activated. The measurement method agreed with the questionnaire in study 1 (Cronbach’s α = 0.87).

#### 4.4.2. Secondary Activation Manipulation

In order to ensure the manipulation effect, stereotype implicit manipulation activation (60 s) should be performed again after completing the questionnaire. The specific process is the same as that in step 3.

#### 4.4.3. Job Performance

In step 4, job performance and intergenerational knowledge transfer intention were measured. In Study 1, job performance is divided into task performance, relationship performance, learning performance, and innovation performance. Considering the experiment duration, practical operation difficulty, and research interest, only innovation performance is selected as a subindex, and “brainstorming task” is used to measure innovation performance. The experiment task was to give three objects, namely, plastic, wire, and wood, and ask the participants to write down all the uses they could think of in two minutes. A previous study [62], measures innovation performance from two dimensions, namely, novelty and fluency. First, according to Amabile’s (1982) evaluation rule, a rater who had no prior knowledge of the experiment content rated the innovation and novelty of the subjects according to his subjective judgment, using a 5-point Likert scale (1 = strongly disagree; 5 = strongly agree) [63]. Second, the rater calculates the number of answer items recorded by the subjects to serve as the basis for innovation fluency. The greater the number, the better the innovation ability. This study has a good consistency (Cronbach’s α = 0.79).

#### 4.4.4. Intergenerational Knowledge Transfer Intention

The “work context task” measured intergenerational knowledge transfer intention. It reads: “Imagine the following situation: After the morning meeting on Monday, you are given a new assignment and need to work with another young colleague on the team. Your task is to plan and implement the company’s annual meeting, but he/she is new and unfamiliar with the business. The work needs to be completed in a week.” The subjects were asked to play a role in this work situation and fill in a 5-item questionnaire according to their ideas using a 5-point Likert scale. The questionnaire questions were as follows: “I will send him the case of the activities planned by the company for his reference and learning”; “I will share my methods/thoughts/workbook with him”, etc. This study has a good consistency (Cronbach’s α = 0.92).

#### 4.4.5. Control Variables

In step 5, subjects fill in the demographic information, including gender, age, organization, occupation, tenure, education level, and health status.

At the end, the experimenter thanked the participants for their participation, explained the research purpose, and rewarded them with a voucher.

### 4.5. Analytical Strategy

For data analysis, SPSS 21.0 was used to test the reliability and validity test. The validity of age stereotype priming was tested by using an independent sample *t*-test. In addition, scale reliability, correlation analysis, regression analysis, and mediation tests were used to conduct hypothesis testing. Bootstrapping with 5000 replications was used to compute the confidence intervals of conditional indirect links.

### 4.6. Research Results

#### 4.6.1. Manipulation Checks

The validity of age stereotypes priming was tested by an independent sample *t*-test. As shown in Table 4, participants’ scores in the positive age stereotypes group in the stereotype priming test items were significantly higher than those in the control group (*t* = 3.497; *p* < 0.01), indicating that implicit activation manipulation was effective.

#### 4.6.2. Hypothesis Testing

##### Testing Main Effect

The *t*-test was used for hypothesis analysis, as shown in Table 4, and it was found that there was a significant difference in novelty between the active group and the negative group (*t* = 6.198; df = 58; *p* < 0.01); that is, the innovation novelty of the active group was significantly higher than the negative group. There was a significant difference in fluency between the active group and the negative group (*t* = 33.491, df = 58, *p* < 0.001); that is, the innovation fluency of the active group was significantly higher than that of the negative group. Let us say that 1a and 1b were verified. There were significant differences in intergenerational knowledge transfer intention between the active and negative groups (*t* = 5.486; df = 58; *p* < 0.01), and hypotheses 2a and 2b were verified. There were significant differences in self-perceived aging between the active group and the negative group (*t* = −4.507, df = 58, *p* < 0.001). Let us say that 3a and 3b are verified.

##### Testing Mediating Effect

Based on controlling five control variables, namely, gender, age, working years, unit type, education level, and health status, self-perceived aging was returned to obtain Model 9. The specific results are shown in Table 5. It can be seen from the data in the table that the age stereotypes have a significant impact on self-perceived aging (β = −0.461, *p* < 0.001), indicating that the age stereotypes have a significant negative impact on self-perceived aging; that is, the more positive the age stereotypes, the lower the degree of self-perceived aging. Model 10 was obtained via regression of innovation novelty. It can be seen from the data in the table that age stereotypes have a significant positive effect on innovation novelty (β = 0.590; *p* < 0.001). The regression of innovation fluency was carried out to obtain Model 12. It can be seen from the data in the table that innovation novelty has a significant impact (β = 0.518; *p* < 0.001), indicating that age stereotypes have a significant positive impact on innovation fluency. Model 14 was obtained by regression of intergenerational knowledge transfer intention. Age stereotypes significantly impacted intergenerational knowledge transfer intention (β = 0.541; *p* < 0.001), indicating that age stereotypes positively impacted intergenerational knowledge transfer intention.

Finally, based on the small-sample mediation analysis method proposed by Preacher and Hayes (2008) [64], this study uses the bootstrap method (sampling repeated 5000 times) to examine the mediating effect of self-perceived aging on the age stereotypes, innovation novelty, innovation fluency, and intergenerational knowledge transfer intention of older employees. Based on Model 9, we add the effect of self-perceived aging on innovation novelty to obtain Model 11; based on Model 10, we add the effect of self-perceived aging on innovation novelty to obtain Model 13; based on Model 12, we add the effect of self-perceived aging on innovation novelty to obtain Model 15.

The results show that after adding self-perceived aging, the effects of age stereotypes on innovation novelty, innovation process, and intergenerational knowledge transfer intention are still significant, indicating that self-perceived aging plays a mediating role in the prediction model of innovation novelty between age stereotypes, and the mediating effect value is 0.57, SE = 0.19, and 95% CI = [0.256, 0.973]. Self-perceived aging has a mediating role in the prediction model of innovation fluency based on age stereotypes. The mediating effect value is 2.89, Boot SE = 0.97, and 95% CI = [1.20, −4.98]. Self-perceived aging played a mediating role in the prediction model of age stereotypes on intergenerational knowledge transfer intention. The mediating effect was 0.38, Boot SE = 0.11, and 95% CI = [0.146, 0.658]. Assume that H3a and H3b are verified.

## 5. Discussion of Research Conclusions

### 5.1. Discussion

This study explores the impact of age stereotypes on the job performance of older employees in the workplace. The results of two studies support that positive age stereotypes have a significant positive impact on job performance, while negative age stereotypes have a significant negative impact on job performance. This is consistent with the conclusion obtained by previous experimental methods [14,21]. Specifically, the recall performance of subjects in the positive age stereotype group in the experimental task was superior to that of individuals both without and with negative stereotype manipulation. At the same time, this study also focuses on the impact of age stereotypes on innovation performance. The results reveal significant differences between the positive and negative groups’ “novelty” and “fluency” dimensions. That is, positive age stereotypes can significantly improve innovation performance. This may be because age does not necessarily hurt job and innovation performance for many knowledge-based older workers. However, although cognitive ability tends to decline with age, the increase in professional knowledge can compensate for the negative impact of such decline [65].

Secondly, older employees who hold positive stereotypes have strong self-actualization motivation, attach importance to learning and development, and are proactive in learning new technologies and methods [5], while learning goals can significantly motivate employees to innovate. Conversely, older employees who hold negative stereotypes doubt the value of older workers in the organization, develop job burnout, and lack motivation to learn new technologies and skills, which can result in knowledge-based older workers’ increasingly outdated knowledge and declining workplace performance with increasing age. The results of both studies support that the positive age stereotypes of older employees significantly positively affect the intergenerational knowledge transfer intention, and the negative age stereotypes of older employees significantly negatively affect the intergenerational knowledge transfer intention. On the one hand, due to their pursuit of life meaning, older employees are more pro-social and willing to help younger colleagues, thereby transferring their own work experience, lessons, and skills [60]; on the other hand, the intention of older employees to transfer knowledge between generations is influenced by the organizational environment. When the organizational environment holds a positive stereotype of older workers, it gives older employees more emotional value to transfer knowledge between generations, which stimulates them to recognize their value in the organization and thus engage in more prosocial behaviors; on the contrary, it will reduce prosocial motivation and the intention to transfer intergenerational knowledge. Therefore, to stimulate the intergenerational knowledge transfer intention of older employees in the organization, the organization needs to focus not only on the personal emotional needs of older employees but also on the affirmation of the organizational environment and atmosphere in terms of their work significance and intergenerational behavior, as this can play a pivotal role in motivating older employees to transfer their knowledge to younger generations.

Both studies support the mediating role of self-perceived aging in age stereotypes and job performance and intergenerational knowledge transfer intention. This is consistent with the conclusion of previous empirical studies that older people with more negative age stereotypes have worse self-perceived aging, and negative self-perceived aging affects life expectancy by affecting survival intention [7]. Aging self-perception is directly related to age stereotypes and acts as a mediating factor in the impact of age stereotypes on cognitive and physical health functioning [31]. At the same time, the research by Levy (2009) in geriatric psychology highlights the importance of learning and internalizing images throughout one’s lifetime [52]. This process assimilates shared age stereotypes into the self-concept of elderly employees, ultimately impacting successful aging in the workplace.

### 5.2. Theoretical Implications

First of all, this study not only broadens the study of age stereotypes in the workplace but also broadens the study of the impact of age stereotypes of age discrimination on successful workplace aging by exploring the mechanism of the impact of aging impressions on older employees’ job performance and intention to transfer cost knowledge. Previous studies on age stereotypes mainly focused on the cognitive ability and psychological and physiological status of retired elderly people [4,15]. By focusing on working 45–65-year-old employees, this study explores the impact of age stereotypes on the work motivation and work behavior of older employees through a combination of questionnaires and experiments. It emphasizes the actual impact of age stereotypes in the workplace, which not only broadens the research scope of age stereotypes. It also provides a new theoretical perspective for organizational behavior research on older employees. Secondly, this study not only focuses on the impact of negative age stereotypes on older employees but also explores the positive effects of positive age stereotypes on the work performance and intergenerational knowledge intention of older employees. The study’s results prove that positive age stereotypes can stimulate the work motivation of older employees, inducing them to be more actively engaged in work and improve work performance. This not only breaks through the limitations of previous organizational behavior studies that focus on negative age stereotypes of older employees [19,40] but also provides a new theoretical basis for organizations to promote the successful aging of older employees from a positive perspective.

Finally, based on age stereotypes, this study enriches the literature on the job performance and intergenerational knowledge transfer intention of older workers. On the one hand, the results support that even with the increase in age, older employees holding positive age stereotypes can still maintain relatively high work performance, providing a new theoretical perspective for improving older employees’ work performance; on the other hand, with the increasing diversity of the age structure of organizations, knowledge transfer between generations becomes increasingly essential [23]. There are few studies on intergenerational knowledge transfer intentions, and it has been found that older employees’ intergenerational knowledge transfer intention is affected by positive and negative age stereotypes. This study provides a new theoretical perspective for organizations to promote the intergenerational knowledge transfer of older employees.

### 5.3. Practical Implications

The study found that positive age stereotypes significantly positively affect older employees’ work performance and intergenerational knowledge transfer intention. This means that to achieve successful workplace aging for older workers, employees must be helped to establish positive age stereotypes. Therefore, organizations should implement age-inclusive human resource management policies, establish an organizational environment for fair development, and provide institutional guarantees for older employees to obtain fair career development, working conditions, and welfare benefits. For example, providing learning and development opportunities for older workers, providing targeted training, focusing on performance feedback from older workers, and providing career planning advice can improve the performance of older workers.

The negative age stereotypes significantly impact older employees’ work performance and intergenerational knowledge transfer intention. This means that organizations must break negative age stereotypes and reduce age discrimination to develop human resources for older workers. This requires organizations to be aware of the harm of negative age stereotypes, to improve employees’ awareness and understanding, and to encourage employees to recognize the impact of stereotypes through training, seminars, and other means of reducing their existence in the organization. In addition, the study also found that age stereotypes influence the workplace behavior of older workers through self-perceived aging. Therefore, for older employees to achieve successful workplace aging, they must also establish a positive self-aging perception, which requires the joint support of organizations and society. Therefore, the organization guides the members of the organization through cultural construction, experience sharing, team cooperation, and other ways; strengthens interaction with the elderly employees; gives recognition and respect to their value in the organization; and improves the positive work significance perception of the elderly employees.

### 5.4. Limitations and Future Research Directions

Although this study explores the application of positive and negative age stereotypes in organizational contexts, the sample types are not comprehensive enough. This limitation introduces potential selection bias, as different occupations have varying requirements for cognitive and physical abilities, which means that age stereotypes may have different effects on the behavioral outcomes of employees. For instance, if the sample predominantly includes knowledge workers, the findings may only partially capture the experiences of non-knowledge workers who face different age-related stereotypes and impacts. Addressing this potential selection bias is crucial for enhancing the validity and applicability of the study’s conclusions. Therefore, future studies should incorporate a more diverse sample, including various types of knowledge and non-knowledge workers across different industries, to better understand the nuanced effects of age stereotypes

The development of the age stereotypes scale originated from the field of psychology, and the content of the measurement is mainly the body, psychology, and cognition of the elderly, which may be inconsistent with concern for elderly workers in the context of enterprises. However, because the current research on age discrimination is still in the preliminary stage and mainly focuses on theoretical exploration, there is still no research on the measurement of age discrimination and age stereotypes in the workplace in China. Therefore, in the future, in the context of China’s aging population, more scale development and related empirical studies on age discrimination and stereotypes in the workplace are needed.

## Figures and Tables

**Figure 1 behavsci-14-00503-f001:**
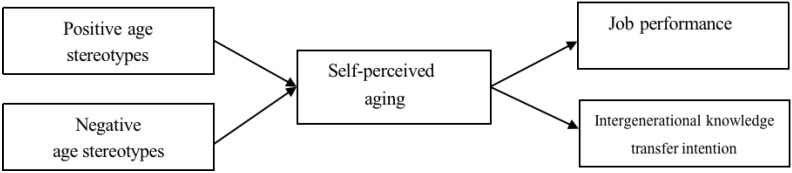
Theoretical framework.

**Table 1 behavsci-14-00503-t001:** Results of confirmatory factor analyses.

Model	χ^2^	df	χ^2^/df	RMSEA	SRMR	CFI	IFI	TLI
Five-factor model	912.925	339	2.693	0.065	0.036	0.916	0.917	0.907
Four-factor model	1490.085	344	4.332	0.091	0.047	0.833	0.834	0.816
Three-factor model	1689.474	347	4.869	0.134	0.098	0.804	0.805	0.787
Two-factor model	1735.378	349	4.972	0.1	0.051	0.798	0.799	0.781
One-factor model	2858.305	350	8.167	0.134	0.063	0.634	0.635	0.605

Note: five-factor model (hypothetical model); four-factor model (combining positive aging stereotypes and negative aging stereotypes into one factor); three-factor model (combining positive aging stereotype, negative aging stereotype, and self-perceived aging into one factor); two-factor model (combining job performance and intergenerational knowledge transfer willingness into one factor); one-factor model (five factors combined into one factor).

**Table 2 behavsci-14-00503-t002:** Means, standard deviations, and correlations among research variables.

Variable	M	SD	1	2	3	4	5	6	7	8	9	10	11	12
1. Positive age stereotypes	3.073	0.451	--	(0.87)										
2. Negative age stereotypes	2.722	0.682	−0.600 **	--	(0.93)									
3. Self-perceived aging	2.530	0.576	−0.604 **	0.603 **	--	(0.94)								
4. Job performance	2.394	0.343	0.595 **	−0.383 **	−0.542 **	--	(0.91)							
5. Intergenerational knowledge transfer intention	4.1	0.635	0.510 **	−0.289 **	−0.511 **	0.771 **	--	(0.83)						
6. Gender	1.52	0.500	−0.153 **	0.378 **	0.123 *	−0.004	0.064	--						
7. Age (years)	1.71	0.868	0.150 **	−0.247 **	−0.149 **	0.014	−0.046	−0.325 **	--					
8. Organization	1.75	0.432	0.183 **	−0.317 **	−0.108 *	0.101 *	0.006	−0.340 **	0.286 **	--				
9. Occupation	1.65	0.478	−0.223 **	0.494 **	0.180 **	−0.046	0.029	0.340 **	−0.304 **	−0.397 **	--			
10. Tenure (years)	2.08	0.523	0.007	−0.059	−0.072	−0.040	−0.039	−0.0150 **	0.514 **	0.032	−0.097	--		
11. Educational	1.49	0.500	0.119 *	−0.217 **	−0.177 **	0.072	0.044	−0.002	0.073	−0.157 **	−0.252 **	0.013	--	
12. Health	1.86	0.352	0.045	−0.038	−0.085	0.018	0.060	−0.039	0.040	−0.022	−0.095	0.022	0.048	--

Note: sample size = 401; * *p* < 0.05, ** *p* < 0.01; Cronbach’s alpha for each study variable is in parentheses on the diagonal.

**Table 3 behavsci-14-00503-t003:** Multilevel regression results.

Variable	Self-Perception of Aging		Job Performance			Intergenerational Knowledge Transfer Intention		
	M1	M2	M3	M4	M5	M6	M7	M8
Gender	0.055	−0.082	0.022	0.064 *	0.049	0.078	0.146 *	0.114
Age (years)	−0.046	−0.008	0.002	−0.02	−0.022	−0.023	−0.061	−0.064
Organization	−0.098	0.036	0.109 *	0.045	0.051	0.097	−0.009	0.005
Occupation	−0.078	0.148 **	−0.016	−0.092 *	−0.065	−0.061	−0.186	−0.127
Tenure (years)	−0.019	−0.063	−0.028	−0.001	−0.012	−0.016	0.028	0.003
Educational	−0.189 **	−0.065	0.068	0.014	0.002	0.083	−0.006	−0.031
Health	−0.11	−0.099	0.019	0.006	−0.012	0.119	0.096	0.057
Positive stereotypes		−0.456 ***		0.415 ***	0.333 ***		0.7 ***	0.521 ***
Negative stereotype		0.391 ***		−0.073 *	−0.002		−0.115 *	0.039
Self-perception of aging					−0.181 ***			−0.392 ***
*R* ^2^	0.268	0.483	0.022	0.38	0.428	0.014	0.306	0.372
Δ*R*^2^	0.268	0.215	0.022	0.358	0.048	0.014	0.292	0.066

Note: * *p* < 0.05; ** *p* < 0.01; *** *p* < 0.001.

**Table 4 behavsci-14-00503-t004:** Results of independent sample *t*-test.

Variable	Positive Stereotypes Groups	Negative Stereotypes Groups	df	*t*
Self-perception of aging	2.45 ± 0.55	3.19 ± 0.7	58	−4.507 ***
Innovation novelty	3.80 ± 0.81	2.47 ± 0.86	58	6.198 **
Innovation fluency	14.0 ± 4.76	8.99 ± 3.36	58	33.491 ***
Intergenerational knowledge transfer intention	3.99 ± 0.61	4.67 ± 0.32	58	5.486 **

Note: ** *p* < 0.01; *** *p* < 0.001.

**Table 5 behavsci-14-00503-t005:** Multilevel regression results.

Variable	Self-Perception of Aging	Job Performance (Innovation Novelty)		Job Performance (Innovation Fluency)		Intergenerational Knowledge Transfer Intention	
	M9	M10	M11	M 12	M13	M14	M15
Gender	−0.158	0.264 *	0.363 *	0.059 **	−0.504	0.109	−0.003
Age(years)	0.124	−0.016	0.014	0.036	0.156	−0.236	−0.021
Organization	−0.242	0.059	−0.018	0.061	−0.115	0.374	0.025
Occupation	−0.143	0.084	0	0.072	−0.112	0.177	0.044
Tenure(years)	0.025	−0.09	−0.162	0.082	1.116	0.026	0.051
Educational	0.084	−0.106	−0.057	−0.044	0.066	0.017	0.107
Age stereotypes	−0.461 ***	0.590 ***	0.677 **	0.518 ***	2.139 *	0.541 ***	0.250 *
Self-perception of aging			−0.859 **		−4.358 ***		−0.569 **
*R* ^2^	0.235	0.421	0.849	0.294	0.7627	0.331	0.865
*F*	3.592 **	7.135 **	16.458 ***	181.887 ***	8.864 ***	5.179 **	18.920 ***

Note: * *p* < 0.05, ** *p* < 0.01, *** *p* < 0.001.

## Data Availability

The datasets generated during and/or analyzed during the current study are available from the corresponding author on reasonable request.

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
