# Peer review of "Effects of Age Stereotypes of Older Workers on Job Performance and Intergenerational Knowledge Transfer Intention and Mediating Mechanisms"

_behavsci, 2024, doi:10.3390/bs14060503_

Round 1
Reviewer 1 Report
Comments and Suggestions for Authors
This paper significantly contributes to our understanding of how age stereotypes affect job performance and intergenerational knowledge transfer among older workers. Notably, the focus on positive age stereotypes alongside the traditionally examined negative stereotypes provides a more nuanced perspective. The use of a mixed-methods approach enhances the depth of the analysis. However, there are several areas where the framing, methodology, and results presentation can be substantially improved.
- While the introduction effectively sets the context, it is surprising that the paper focuses on older employees' own stereotypes rather than those held by employers, especially because the paper looks at work-related outcomes. Elaborating on the rationale for this choice (i.e. why are not employers’ stereotypes relevant in this specific study), would strengthen the argument.
- The decision to include older workers aged 45-65 also requires further justification. While the lower limit is supported by existing literature, the rationale for the upper limit of 65 is unclear and needs clarification. Many people at least in Europe work beyond 65 and also later.
- Page 3: The discussion on different age stereotypes do not consider the fact that older workers can score differently on different types of skills. For example, previous literature on age stereotypes have made a distinguishment between soft and hard skills, where soft skills refer to organizational behaviour such as loyalty, whereas hard skills refer to productivity, learning new things etc., see eg. Retention of Older Workers: Impact of Managers’ Age Norms and Stereotypes | European Sociological Review | Oxford Academic (oup.com) Previous research also find that older workers are judge harder on hard skills than soft skills. Do you think this might affect your results?
- Page 4, Line 168: Clarify which empirical study is being referred to. Is it the study itself or another study?
- Page 5. The repeated use of "on the one hand" without a corresponding "on the other hand" is confusing. This phrasing should be revised for clarity.
- The distinction between age stereotypes and self-perceived aging as mediators needs clearer explanation. How do older employees age stereotypes about older employees differ from self-perceived aging, and why is this distinction important? Or asked in another way: What is “left” unexplained in the age stereotype variables after self-perceived ageing is included in the model. How do we interpret that?
- Page 7: The methods section needs more clarity, particularly regarding the items used to measure age stereotypes. While "patient" and "creative" can be considered soft skills cf. my previous comment, there are also other characteristics/skills. Does the index measure different types of skills?
- Clarify the purpose of the confirmatory factor analysis, as it is unclear why the analysis section starts with this. Is it to test for multicollinearity between variables?
- The introduction of control variables is crucial, but the interpretation of coefficients in Table 3 is unclear due to the absence of reference categories.
- Page 10, Section 3.4.2: The introduction of Table 4 appears misplaced as it relates to Study 2. Ensure all tables are correctly referenced and placed.
- Page 11, Line 441: The sentence is incomplete or improperly constructed. Revise for clarity.
- Table 5: Improve readability by listing the name of the dependent variables rather than using A, B, C, etc.
- The experimental procedure on page 11 lacks sufficient detail. How were participants manipulated? Did they observe age stereotypes directly?
- The discussion effectively interprets the findings and links them back to the hypotheses. However, further elaboration on potential selection bias page 16, line 665, and its implications would enhance the discussion, i.e. how might selection bias affect the results?
Comments on the Quality of English LanguageIt needs to be proofread
Reviewer 2 Report
Comments and Suggestions for Authors
Congratulations on the work done on the "effects of age stereotypes of older workers on job performance and intergenerational knowledge transfer intention and mediating mechanisms, understanding age stereotypes and their impact on older workers in organizational contexts". The topic is relevant in today's workforce, especially considering the aging population.
The work is well-grounded in relevant literature, based on theoretical constructs with appropriate conceptual foundations. It is well-structured, with sections written to explicitly outline the required content for each section, in clear and understandable language. However, some minor adjustments could be made, particularly in the method section:
In study 1, there is a lack of a point indicating the procedures adopted in the data analysis, instead of this reference being made in the results section of study 1. In section 3.1, it only refers to the sample, and it could include the title sample and procedures, as these are described in this section. Additionally, the authors mention that rewards were given to participants, and it could be indicated what these were.
In study 2, it is mentioned that it was conducted offline, but it was not clear whether it was conducted in person and how, which I believe could be better explained. Similarly, in study 2, there could be a section outlining the procedures adopted in the data analysis, instead of this reference being made in the results section of study 2.
Round 2
Reviewer 1 Report
Comments and Suggestions for Authors
I believe the authors adressed all comments properly. However, there are some minor issues:
- Comment and response to #13. "... and the proportion of enterprises is 48... It is three percent"? Something is missing in that sentence or?
- Also response to #13.There are still some issues on the descriptions of the experiment that I do not follow. For example it says that Study 2 was conducted offline. What does that mean? I mainly get confused because it is described later that the researcher was present?
